# Identification of Dihydromyricetin and Metabolites in Serum and Brain Associated with Acute Anti-Ethanol Intoxicating Effects in Mice

**DOI:** 10.3390/ijms22147460

**Published:** 2021-07-12

**Authors:** Eileen Carry, Dushyant Kshatriya, Joshua Silva, Daryl L. Davies, Bo Yuan, Qingli Wu, Harna Patel, Elizabeth R. Park, John Gilleran, Lihong Hao, Jacques Roberge, Nicholas T. Bello, James E. Simon

**Affiliations:** 1Department of Medicinal Chemistry, Ernest Mario School of Pharmacy, Rutgers University, Piscataway, NJ 08854, USA; emc139@rutgers.edu (E.C.); qlwu@sebs.rutgers.edu (Q.W.); 2New Use Agriculture & Natural Plant Products Program, Department of Plant Biology, School of Environmental and Biological Sciences, Rutgers University, New Brunswick, NJ 08901, USA; bo_yuan@hsph.harvard.edu (B.Y.); hkp38@scarletmail.rutgers.edu (H.P.); 3Department of Molecular Design and Synthesis, Office of Research and Economic Development, Rutgers University, Piscataway, NJ 08854, USA; erpark@stanford.edu (E.R.P.); jg1136@ored.rutgers.edu (J.G.); jr1257@research.rutgers.edu (J.R.); 4Department of Animal Sciences, School of Environmental and Biological Sciences, Rutgers University, New Brunswick, NJ 08901, USA; dsk118@scarletmail.rutgers.edu (D.K.); haoli@sebs.rutgers.edu (L.H.); 5Titus Family Department of Clinical Pharmacy, School of Pharmacy, University Southern California, Los Angeles, CA 90089, USA; silvajos@usc.edu (J.S.); ddavies@usc.edu (D.L.D.)

**Keywords:** dihydromyricetin, metabolism, bioavailability, GABA_A_ receptors, acute alcohol intoxication, alcohol use disorder, loss of righting reflex

## Abstract

Dihydromyricetin is a natural bioactive flavonoid with unique GABA_A_ receptor activity with a putative mechanism of action to reduce the intoxication effects of ethanol. Although dihydromyricetin’s poor oral bioavailability limits clinical utility, the promise of this mechanism for the treatment of alcohol use disorder warrants further investigation into its specificity and druggable potential. These experiments investigated the bioavailability of dihydromyricetin in the brain and serum associated with acute anti-intoxicating effects in C57BL/6J mice. Dihydromyricetin (50 mg/kg IP) administered 0 or 15-min prior to ethanol (PO 5 g/kg) significantly reduced ethanol-induced loss of righting reflex. Total serum exposures (AUC0_→_24) of dihydromyricetin (PO 50 mg/kg) via oral (PO) administration were determined to be 2.5 µM × h (male) and 0.7 µM × h (female), while intraperitoneal (IP) administration led to 23.8-fold and 7.2- increases in AUC0_→_24 in male and female mice, respectively. Electrophysiology studies in α5β3γ2 GABA_A_ receptors expressed in Xenopus oocytes suggest dihydromyricetin (10 µM) potentiates GABAergic activity (+43.2%), and the metabolite 4-O-methyl-dihydromyricetin (10 µM) negatively modulates GABAergic activity (−12.6%). Our results indicate that administration route and sex significantly impact DHM bioavailability in mice, which is limited by poor absorption and rapid clearance. This correlates with the observed short duration of DHM’s anti-intoxicating properties and highlights the need for further investigation into mechanism of DHM’s potential anti-intoxicating properties.

## 1. Introduction

Alcohol use disorder (AUD) is the most common substance use disorder (SUD), responsible for more than 3 million deaths each year [1]. Current AUD pharmacotherapies have demonstrated limited clinical efficacy, with drug discovery efforts hindered by ambiguity of relevant molecular targets [2,3]. Although the utility of pharmacotherapies for treatment of addictive disorders has been debated, the more recent efficacy of pharmacotherapies for opiate use disorder (OUD) supports their utility [4,5,6]. This may be relevant to AUD medication development when considering the similarities in inhibitory neurological effects of opiates and alcohol, both of which can cause fatal respiratory depression at toxic doses [7,8]. Recognition of the µ-opioid receptor as the major molecular target of opiates, has enabled the development of OUD pharmacotherapies, such as naltrexone and naloxone, µ-opioid receptor antagonists that counteracts inhibitory neurological effects of opiates. These medications have demonstrated substantial clinical efficacy in improving OUD recovery success rates and mitigating opiate related overdose fatalities [4,9]. This suggests comparable treatment options for AUD, such as those that counteract major inhibitory neurological effects of ethanol, may provide utility in the treatment of AUD and clinically relevant acute alcohol intoxication (AAI).

While u-opiate receptors are well-established as the molecular target of opiates, the major molecular target (s) of ethanol is less clear. However, decades of research indicates that GABA_A_R-mediated responses may play a dominant role in mediating the neurological effects of ethanol [10,11,12,13,14]. GABA_A_Rs are the major inhibitory neurotransmitter in the brain and are a common target of anti-anxiety medications, including benzodiazepines (BZDs) [15]. Both ethanol and BZDs exert additive intoxicating effects that lead to cross tolerance to other GABAergic drugs [10,11,12,16,17,18]. This is consistent with overlapping activation of the same neuroreceptor. In fact, comparable to the use of opiate agonists for opiate withdrawal, BZDs are an effective first line of treatment for alcohol withdrawal [19,20]. Collectively, this supports the major role that GABA_A_Rs play in mediating many of the behavioral effects of ethanol. 

Dihydromyricetin (DHM) is a natural flavonoid that could provide a link to targeting inhibition of ethanol-induced GABA_A_R potentiation to counteract major neurological effects of ethanol. DHM is currently sold as a dietary supplement on the US market and has traditionally been used in Asian medicine to naturally counteract ethanol intoxication and prevent hangovers [21]. More recently, preclinical studies in rats demonstrated DHM to substantially reduce ethanol-induced loss of righting reflex (LORR), subsequent withdrawal symptoms, ethanol-induced GABA_A_R plasticity, and prevent physiological symptoms of fetal alcohol syndrome [22,23,24]. Evidence suggests that the mechanism of the anti-ethanol properties of DHM may be linked to specific GABA_A_R molecular interactions at the BZD binding site. Electrophysiology studies indicate DHM acts as a GABA_A_R positive modulator and inhibitor of ethanol-induced GABA_A_R potentiation. Importantly, both ex vivo and in vivo anti-ethanol effects of DHM were antagonized dose-dependently by the addition of flumazenil, a BZD antagonist [22,23]. These studies support the potential of DHM bioactivity in counteracting major neurological effects of ethanol through specific GABA_A_R interactions at the BZD binding site.

While DHM’s bioactivity is promising, a major obstacle to its clinical efficacy is its suboptimal pharmacokinetic (PK) properties, especially when targeting CNS effects. As with most flavonoids, DHM is poorly absorbed into the blood stream when taken orally, with a short, variable half-life that is not consistent with reliable clinical efficacy. For instance, literature reports that in rats, DHM had an oral absolute bioavailability of only 4.02% [25]. Thus, obtaining reliable and adequate CNS exposure is a major hinderance to translatable clinical efficacy. The poor bioavailability of DHM is linked to the multiple hydroxyl substituents resulting in a high polar surface area (PSA = 147), well outside the ideal for drug candidates. Current literature suggests that compounds with a PSA > 140 have poor absorption (<10%) [26] and a PSA < 70 are ideal for blood–brain barrier (BBB) penetration [27]. The hydroxyl substituents are also sites of metabolic instability, as they are rapidly conjugated through phase II metabolism [28]. Thus, while the pharmacological bioactivity of DHM is promising, structural modifications are necessary to increase druglikeness, and thus potential for improved bioavailability and CNS exposure.

To enhance bioavailability, multiple in vivo efficacy studies with DHM have utilized intraperitoneal (IP) administration, including those demonstrating anti-intoxicating effects of ethanol [22,29]. However, there are no reports of DHM bioavailability with IP administration, necessary for comparison of in vivo efficacy with serum and tissue exposure to DHM. As DHM is a polyhydroxylated flavonoid, which are often associated with non-specific bioactivities and bioactive metabolites that contribute to these effects [30], it is especially important to establish a clear relationship with DHM exposure levels and acute anti-intoxicating effects, as well as explore potential bioactive metabolites. Although studies have identified DHM in fecal and urine samples, to date, no studies have established serum and brain exposure to DHM metabolites, important to assess potential for bioactivity.

We report the first investigation into serum and brain exposure to DHM and metabolites associated with acute anti-intoxicating properties in mice. First, we quantified DHM and identified metabolites in serum and brain samples following oral (PO) and IP administration. Next, we conducted the first efficacy study in mice to assess the time course of DHM’s acute anti-intoxicating properties at the high ethanol dose of 5 g/kg PO. Lastly, we synthesized three DHM metabolites, 3′-O-methyl-dihydromyricetin (3′-Me-DHM), 4′-O-methyl-dihydromyricetin (4′-Me-DHM) and 4′-dehydroxy-dihydromyricetin (4′-DeOH-DHM), enabling preliminary screening for GABAergic potentiation in Xenopus oocytes.

## 2. Results

### 2.1. Ethanol-Induced Loss of Righting Reflex

Ethanol-induced loss of righting reflex (LORR) was assessed in male and female mice (*n* = 8) following ethanol PO with and without DHM IP 50 mg/kg administration at 0, 15, 45, 90, and 180 min prior to ethanol administration. There was an effect for sex (F (1, 11) = 7.5, *p* < 0.05) and time for dose (F (5, 55) = 5.9, *p* < 0.0001). Females demonstrated reduced LORR (*p* < 0.05). Male and females were separately analyzed (Figure 1A,B). Males demonstrated an effect for DHM (F (5, 25) = 2.7, *p* < 0.05), but there was no difference between time of DHM injection. Females also demonstrated an effect for DHM (F (5, 30) = 3.8, *p* < 0.005), with DHM injected at time 0 and 45 min before oral ethanol reducing LORR compared with vehicle treated mice (*p* < 0.05). There was a significant reduction in LORR in mice (*n* = 8; 3 males/5 females) demonstrating tolerance between Baseline (week 1) to Post (week 8) (t (7) = −5.39, *p* < 0.005), whereas there was no difference in LORR in responsive mice (*n* = 8; 5 males/3 females; (t (7) = 0.3, *p* = 0.97)) (Figure 1C). For responsive mice, there was an effect of DHM (F (5, 25) = 4.3, *p* <0.001) with DHM injected at time 0 and time 15 prior to ethanol PO demonstrated a reduction in LORR (*p* < 0.05) (Figure 1D).

### 2.2. Quantification of DHM in Serum and Brain Samples

The C_max_ and AUC0_→_24 were calculated based on the determined serum concentrations of DHM at 8 time points (0, 0.25, 0.75, 1.5, 3, 6, 12 and 24 h) following both IP and PO administration (*n* = 3–4, per sex and time point) (Table 1, Figure 2). The IP C_max_ was 23.8 µM in female mice and 38.3 µM in male mice. Following PO administration of DHM, the C_max_ was 1.9 µM and 0.62 µM in male and female mice, respectively. The factorial analysis for the DHM concentrations at 15 min (the C_max_) revealed that there were effects for sex (F (1,8) = 11.9, *p* < 0.01), route of administration (F (1, 8) = 101.4, *p* < 0.0001) and sex X route of administration (F (1, 8) = 8.7, *p* < 0.05). Post hoc testing revealed that at 15 min, male mice via IP administration had the highest serum concentrations of DHM (*p* < 0.001). At 45 min, there were effects for sex (F (1, 9) = 15.5, *p* < 0.005), route of administration (F (1, 9) = 138.8, *p* < 0.00001), and sex X route of administration (F (1, 9) = 25.3, *p* < 0.005). Post hoc testing revealed that at 45 min, female mice via IP administration had significantly greater serum concentrations of DHM as compared to male mice (*p* < 0.001). Despite statistical differences in IP C_max_ and C_45 min_, there was no statistical difference in the IP AUC0_→_24 between male (18.1 µM × h) and female mice (17.3 µM × h). The PO AUC0_→_24 was 2.51 and 0.728 µM × h in male and female mice, respectively. The differences in PO AUC0_→_24 between sexes was statistically significant (*p* < 0.001). A substantial difference was observed between the IP and PO AUC0_→_24 in both sexes of mice. In male mice, the IP AUC0_→_24 was 7.2 times greater than that of the PO AUC0_→_24 (*p* < 0.001). In female mice, the IP AUC0_→_24 was 23.8-fold the PO AUC0_→_24 (*p* < 0.001).

Dihydromyricetin was detected in brain tissue following both PO and IP administration of DHM 50 mg/kg IP at 15 min post DHM administration. DHM was not detected in brain samples 45 min or longer post PO or IP administration. Concentrations were in the range of 50.9–861.5 nmol/g brain tissue.

### 2.3. Identification of Dihydromyricetin Metabolites

Metabolite identification was achieved based on the MRM mass transitions of predicted DHM metabolites (see Appendix A for mass transitions and retention times). The proposed metabolic pathway of DHM, based on literature and those detected in this study, are illustrated in Figure 3 [31,32]. Using a newly developed UPLC-QqQ/MS method, a total of 7 metabolites were successfully detected in serum following both IP and PO administration. These metabolites were identified as glucuronide-dihydromyricetin (G-DHM), 3′-O-methyl-dihydromyricetin, 4′-Me-DHM, 4′-DeOH-DHM), glucuronide-3′-O-methoxy-dihydromyricetin (G-3′-Me-DHM), glucuronide-3′-O-methoxy-dihydromyriceitn (G-4′-Me-DHM) and glucuronide-dehydroxy-dihydromyricetin (G-DeOH-DHM). Except for sulfate-DHM and 4-hydroxy-dihydromyricetin, which were not detected in serum samples, the metabolites identified in this study are in agreement with those previously detected in feces or urine samples [31,32]. Two O-methyl metabolites (3′-Me-DHM and 4′-Me-DHM) were detected at trace levels in brain samples of in male and female mice 15 min post DHM IP 50 mg/kg.

### 2.4. Intrinsic Activity in α5β3γ2 GABA_A_ Receptors Expressed in Xenopus Oocytes

Electrophysiology studies to assess intrinsic α5β3γ2 GABA_A_R activity in Xenopus oocytes were conducted with and without 10 µM concentrations of DHM, 3′-Me-DHM and 4′-Me-DHM (Table 2). DHM significantly potentiated GABA_A_Rs exposed to GABA by +43.2% (*p* < 0.05), in comparison to weekly active, 4′-DeOH-DHM, which potentiated GABA_A_Rs by +5.2%. The activity of 4′-DeOH-DHM did not reach significance (*p* = 0.06). 4′-Me-DHM negatively modulated GABA_A_R potentiation by −12.6% (*p* < 0.05) and 3′-Me-DHM was not active.

## 3. Discussion

Dihydromyricetin (DHM) is a flavonoid with properties that reduce the detrimental impairments associated with ethanol intoxication. Several aspects of dose-related effects of DHM were not fully characterized. With IP administration being a common administration route of DHM efficacy studies, including those demonstrating anti-intoxicating properties, we investigated serum and brain exposure to DHM following PO and IP administration. We found that IP administration of DHM led to significantly increased serum concentrations as compared to PO administration. One factor contributing to the poor oral bioavailability of DHM may be its poor cellular permeability, which was reported as 1.84 × 10^−6^ cm/s [33]. Another factor limiting the oral bioavailability of DHM may be its instability under the mildly alkaline conditions of the lower GI tract. Xiang et al. demonstrated that DHM is stable in acidic environments, similar to that of the upper GI tract, but is unstable in solutions with pH > 6 [34]. Thus, substantial degradation of DHM is expected in the lower GI tract [34]. Additionally, Huang et al. demonstrated that absorption of DHM was increased by the addition of verapamil, an inhibitor of P-glycoprotein (Pgp) efflux transporters [35]. This suggests that activity of Pgp transporters in the intestinal tract decreases DHM bioavailability and correlates with reports of DHM to act as a Pgp substate and inhibitor [36,37]. Interestingly, we found sex to have a significant influence on DHM serum exposure following PO, but not IP administration. As prior studies suggest that sex and certain hormones can impact Pgp expression [38,39], variations in Pgp expression between sexes might contribute to differences in DHM PO exposure. One limitation of our studies was that we did not power our study to perform vaginal cytology to determine the stage of estrous, which could have influenced bioavailability. 

We identified a total of seven DHM metabolites in serum using UPLC-QqQ-MS/MS analysis, enabling the first in vivo identification of DHM metabolites in serum samples. The major metabolic pathways included O-methylation, glucuronidation and dehydroxylation. Metabolites detected in serum samples included 3′-Me-DHM, 4′-Me-DHM and 4′-DeOH-DHM and the corresponding glucuronide conjugates, G-DHM, G-3′-Me-DHM, G-4′-Me-DHM and G-4′-DeOH-DHM. This is consistent with metabolite identification by both Zhang et al. and Fan et al. in urine and fecal samples. However, while 4′-DeOH-DHM was the major metabolite detected in fecal samples [31,32], we only detected 4′-DeOH-DHM at trace levels in serum samples. This correlates with reports suggesting 4′DeOH-DHM to be produced by bacterial fermentation in the large intestines [40]. The high fecal concentration of this 4′-DeOH-DHM, coupled with minimal serum levels, correlates with this metabolite being produced in the large intestines by bacterial fermentation [40] but is poorly absorbed into the blood stream. 

As brain exposure is necessary for GABA_A_R activity, we analyzed brain tissue samples for DHM and metabolites utilizing UPLC-QqQ-MS/MS analysis. We detected DHM 15 min post PO and IP administration of DHM, but not at 45 min or later. As brain exposure to xenobiotics is dependent on blood levels, this correlates with the observed rapid serum clearance. Further, as Pgp transport plays a significant role in the efflux of xenobiotics from the BBB, it is likely that Pgp efflux contributes to the rapid brain clearance of DHM. The detection of DHM in brain tissue is in agreement with findings by Fen et al., who report DHM in rat brain tissue post 100 mg/kg PO [31]. However, Fen et al. detected DHM in brain tissue of Sprague Dawley rats up to 12 h post DHM administration, while we only detected DHM in mouse brain tissue 15 min post DHM administration, and not later. The discrepancy in the time course of brain exposure to DHM could be related to the difference in rodent species, as substantial variations are commonly observed in xenobiotic bioavailability among rats and mice. It is also important to note that brain tissue levels do not accurately represent free extracellular levels of xenobiotic, necessary to exert effect. Thus, future analysis of brain extracellular fluid using tissue microdialysis could provide for a more accurate representation of in vivo GABA_A_R exposure to DHM. Notably, we also detected 3′-Me-DHM and 4′-Me-DHM in brain tissue of male and female mice 15 min post IP, but not PO administration of DHM, although at trace levels. As O-methylated flavonoids have been shown to exhibit more favorable PK properties, with improved GI absorption and stability [40], bioactive O-methyl DHM metabolite (s) could be of significance.

As a preliminary screening for GABAergic activity, we assessed DHM and 3 metabolites (10 µM) for potentiation of α5β3γ2 GABA_A_Rs expressed in Xenopus oocytes. GABA_A_R α5β3γ2 subtypes were selected because of reliability of expression in oocytes, and high homology among subtypes. We observed significant positive modulation by DHM (+43.2%), negative modulation by 3′-Me-DHM (−12.6%). It is also important to note that 3′-Me-DHM, 4′-Me-DHM and 4′-DeOH-DHM were all in the racemic, trans R,R and S,S enantiomeric forms, while DHM, isolated from plant material, was in the enantiomerically pure R,R form. As enantiomeric purity of similar flavonoids to DHM has been shown to influence GABA_A_R activity, this could have diluted effects observed by synthesized racemic metabolites in comparison to the activity of enantiomerically pure DHM. While the observed in vitro GABAergic activity of DHM and 3′-Me-DHM is significant, it is not conclusive of in vivo GABA_A_R activity. Since this activity is limited to a 10 µM concentration and we did not determine free extracellular brain levels, we cannot conclude that compounds reach levels consistent with the reported GABAergic activity. This is a limitation of this study that highlights the need for further investigations into GABAergic activity of DHM and metabolites.

While interpretation of the reported GABAergic activity is limited, previous studies support the reliance of DHM’s anti-intoxicating properties on GABAergic activity. For instance, in rats, the anti-alcohol effects of DHM were antagonized by flumazenil, a BZD antagonist, in a dose-dependent manner [22]. This strongly supports the reliance of DHM’s acute anti-intoxicating properties on specific GABA_A_R interactions in vivo. There are currently no previous reports of GABAergic activity of 3′-Me-DHM, suggesting a need for further investigation. However, it is improbable is that 3′-Me-DHM plays a major role in a GABA_A_R-mediated mechanism of DHM, as methylated metabolites were only minimally detected in brain tissue. Thus, unless extremely potent, it is unlikely that 3′-Me-DHM reaches levels necessary to exert prolonged GABAergic activity. Notably, the selective GABA_A_R positive modulation by DHM correlates with reported SAR studies surrounding flavone GABA_A_R activity that suggest minor molecular modifications significantly influence intrinsic GABA_A_R activity [41,42,43,44]. This further supports the specificity of DHM’s GABA_A_R activity, necessitating further investigation to more fully characterize the GABAergic activity of DHM and metabolites. 

While previous studies have demonstrated anti-intoxicating effects of DHM in rats, we are the first to report a significant reduction in ethanol-induced LORR in mice. Our results indicate that the acute anti-intoxicating effects of DHM may be limited by rapid serum and brain clearance of DHM. Interestingly, we observed a significant response to DHM only at 0, 45 versus 0, 15 min prior to ethanol administration in female and male mice, respectively. This may be related to female mice exhibiting a significantly higher DHM serum concentration 45 min post DHM IP over that of male mice. However, it should be recognized that time points of DHM efficacy are complicated by expected variability in blood ethanol concentrations (BEC). Additionally, evidence indicates that DHM may reduce BEC through activation of ethanol metabolizing enzymes [22,45]. Enhanced ethanol metabolism is a property common of many flavonoids related to non-specific activity shared by polyphenolic structures [46,47]. In contrast to this, inhibition of ethanol-induced GABA_A_R potentiation and acute anti-intoxicating properties are not common to polyphenolic flavonoids, and thus unique to DHM [23]. This is consistent with the reliance of DHM’s GABA_A_R modulation and anti-intoxicating properties on specific, molecular interactions. 

## 4. Materials and Methods

### 4.1. Materials

DHM, isoquercetin, and ascorbic acid were obtained from Sigma-Aldridge (St. Louis, MO, USA). Formic acid, acetonitrile (ACN), methanol (MeOH), hydrochloric acid (HCl) and ethyl acetate (EtOAc) were obtained from Fisher Scientific (Pittsburgh, PA, USA), and Pierce™ LC-MS water from Thermo Fisher Scientific (Waltham, MA, USA). Synthetic precursors, including 3,4-dihydroxy-5-methoxybenzaldehyde, 3,4,5-trihydroxybenzaldehyde and 3,5-benzyloxybenzaldehyde were purchased from Combi-Blocks (San Diego, CA, USA). All synthetic precursors and natural products were ≥95% pure.

### 4.2. Animal Studies

Eight-week-old male and female C57BL/6J mice (Jackson’s Laboratories, Bar Harbor, ME, USA) were housed 4 mice to a cage (separated by sex) and maintained on 12-h light, 12-h dark cycles (lights on at 0700 h) in the animal facility at the Department of Animal Sciences, Rutgers University. Following 1-week acclimation, animals were given ad libitum access to water and a polyphenol-free diet (PFD; 3.82 kcal/g, 10% fat, 20% protein, 70% carbohydrate; D12450H; Research Diets, Inc., New Brunswick, NJ, USA) for at least 10 days. For PK studies, after an overnight fast, mice were administered 50 mg/kg DHM (dissolved in 0.9% saline with 10% DMSO) by either IP injection or oral gavage. Oral gavage was performed using single-use, sterile plastic feeding tubes for mice (20 ga × 30 mm; cat# FTP-20-30, Instech Laboratories, (Plymouth Meeting, PA, USA). Mice were deeply anesthetized with 5% isoflurane mixed with oxygen, and blood was collected by cardiac puncture. Following cardiac puncture, animals were exsanguinated and transcardial perfusion with 0.9% saline was performed to ensure blood removal from the brain tissue. Blood and brain samples were collected from a total of 85 mice at 0.25, 0.75, 1.5, 3, 6, 12, 24 h (3–4 mice for each sex and timepoint). Tissues were immediately homogenized with 0.2% formic acid at 1:2 (*w*/*v*), and frozen at −80 °C until analysis. Blood was centrifuged at 3000× *g* for 10 min at 4 °C to isolate serum. For efficacy studies, mice (*n* = 8/sex) were fasted at 5 h prior to Ethanol gavage 5 g/kg (37% Ethanol in water). DHM 50 mg/kg IP (dissolved in 0.9% saline with 10% DMSO) was administered at 0, 15, 45, 90 and 180 min prior to Ethanol oral gavage. Each mouse was tested at each time point 7 days apart over 6-week period. At the time of Ethanol gavage, mice were placed in oversized conical holders (L: 152 mm X W: 37 mm) and observed for ability to return upright upon being placed in a supine position. Duration of LORR was determined by ability to return upright 3 times within 30 s [48]. To determine whether ethanol-induced tolerance developed, one week (Week 1; Baseline) prior to the 6-week repeated DHM dosing study all mice were oral dose with ethanol (5 g/kg) without any IP injection and LORR was measured. Mice were then re-tested in a similar fashion with oral ethanol 1 week after the DHM 6-week period (Week 8; Post). Mice that demonstrated a significant reduction in LORR from Week 1 to Week 8 were considered “tolerant”, whereas mice that did not demonstrate a significant reduction in LORR from Baseline (Week 1) to Post (Week 8) were considered “responsive”. All animal studies were approved by the Institutional Animal Care and Use Committee of Rutgers University (Bello; PROTO999900014, OLAW #A3262-01, 15 March 2017) and complied with NIH Guide for the Care and Use of Laboratory Animals.

### 4.3. Bioanalysis of Serum and Brain Samples

All serum samples were stored at −80 °C and conditioned to room temperature prior to analysis. An aliquot of 100 µL of serum, 20 µL 4M HCl, and 50 µL of internal standard (IS) solution (417 ng/mL isoquercetin in EtOAc) were combined in an Eppendorf tube. An aliquot of 500 µL of EtOAc was added and samples were vortexed for 30 s. Samples were then centrifuged at 3000× *g* for 2 min and the supernatant transferred to a glass vial. Samples were extracted 2 more times with EtOAc and the supernatant added to the corresponding vial containing 10 µL of 2% ascorbic acid in methanol (MeOH). Samples were then dried under vacuum, reconstituted with 100 µL of 45% MeOH in water with 0.1% formic acid (FA), centrifuged at 16,000× *g* for 10 min and transferred to an HPLC vial for analysis.

Whole brain samples were homogenized in FA. All brain homogenates were stored at −80 °C and conditioned to room temperature prior to analysis. An aliquot of 300 µL of brain homogenate was added to 50 µL IS solution (417 ng/mL in EtOAc) and 500 µL of EtOAc in an Eppendorf tube. Samples were then vortexed for 30 s and sonicated for 1 min. Samples were centrifuged for 2 min at 3000× *g* and the supernatant transferred to a glass vial with 10 µL of 2% ascorbic acid in methanol. Samples were extracted two more times with 500 µL EtOAc and the supernatant transferred to the corresponding vial and dried under vacuum. Each sample was then reconstituted with 100 µL of 45% MeOH with 0.1% FA, centrifuged, and aliquoted into an HPLC vial for analysis.

### 4.4. Instrumentation and Conditions

An Agilent 1290 Infinity II UPLC system interfaced with an Agilent 6470 Triple Quadrupole Mass Spectrometer with an electrospray ionization source (Agilent Technology, Palo Alto, CA, USA) was used for sample analysis. An Waters Acquity UPLC BEH C8 column (2.1 × 150 mm, 1.7 μm) (Milford, MA, USA) column with a VanGuard Acquity C8 guard column (2.1 × 5 mm, 1.7 μm) (Milford, MA, USA) was used for chromatographic separation. The binary mobile phases consisted of water with 0.1% FA (phase A) and acetonitrile (ACN) with 0.1% formic acid (phase B). The flow rate was set at 0.450 mL/min. The column temperature was set to 30 °C and the autosampler to 4 °C. The LC gradient program for each run started at 3% (B%), and was held for 1.5 min before increasing to 22% in 4.0 min, to 40% in 2 min. The column was equilibrated for 2 min before the next injection. An injection volume of 3 μL was used for all standards and samples analyzed. Mass spectral data acquisition was achieved at negative polarity (ESI-). Mass transitions were monitored over a period of 6.5 min with a dwell time of 200 ms. The multiplier voltage for all analytes were at −350 V, the ESI capillary voltage at −3.0 kV, nozzle voltage at −1.5 kV, nebulizer gas (N_2_) pressure at 30 psi, dry gas temperature at 300 °C with a flow rate of 13.0 L/min and sheath gas temperature at 200 °C with a flow rate of 12.0 L/min.

### 4.5. Identification of DHM and Metabolites in Serum and Brain Samples

DHM was identified in serum and brain samples by comparison of retention time and multiple reaction monitoring (MRM) transitions of DHM standard. Expected mass transitions of the following metabolites were included in the analytical method; 4-hydroxy-dihydromyricetin (4-OH-DHM), 3′-deyhdroxy-dihydromyricetin (3′-DeOH-DHM), 4′-dehydroxy-dihydromyricetin (4′-DeOH-DHM), 5-dehydroxy-dihydromyricetin (5-DeOH-DHM), 7-dehydroxydihydromyricetin (7-DeOH-DHM), 5-methoxy-dihydromyricetin (5-Me-DHM), 7-methoxy-dihydromyricetin (7-Me-DHM), 3′-methoxy-dihydromyricetin (3′-Me-DHM), 4′-methoxydihydromyricetin (4′-Me-DHM), dihydromyricetin sulfate, dihydromyricetin glucuronide (G-DHM), dihydroxy-dihydromyricetin glucuronide (G-DeOH-DHM), methoxy-dihydromyricetin glucuronide (G-Me-DHM) and 4-hydroxy-dihydromyricetin glucuronide (G-4-OH-DHM) (see Appendix A for mass transitions). The mass transitions we observed were consistent with those previously reported by Fan et al., 2017 and, in a corresponding manner, were used to predict MRM transitions of proposed DHM metabolites (see Appendix A). DHM was quantified using a calibration curve of DHM at 12 concentrations (0.88–2800 ng/mL). Isoquercetin was used as the internal standard (IS) at a final concentration of 209 ng/mL in all samples and calibration standards. 

### 4.6. Synthesis and Identification of DHM Metabolites

Synthesis of 3′-Me-DHM, 4′-Me-DHM and 4′-DeOH-DHM were completed following the same synthetic route, as depicted in Figure 4. The identity and purity of all compounds were determined using LC-MS, ^1^H-NMR and ^13^C-NMR analysis. All final synthetic products were ≥95% pure. ^1^H-NMR, and ^13^C-NMR analysis were conducted on a Varian VNMRS 500 MHz or a Varian VNMRS 300 MHz. For detailed description of synthetic steps, isolation, and structural characterization please refer to Appendix A.

### 4.7. Xenopus Oocytes Preparation

Xenopus oocytes were stored in incubation media (pH 7.5), consisting of ND96 supplemented with 2 mM sodium pyruvate, 0.1 nM gentamycin, and 10 mL heat inactivated HyClone horse serum. Stage 4 to 5 oocytes were injected with 40 nL of cDNA coding for α5, β3, and γ2 subunits in a ratio of 1:1:10, utilizing a Drummond Nanoject III. Injected oocytes were stored at 18 °C and used in electrophysiology studies within 2–7 days of injection.

### 4.8. Electrophysiology Studies

Whole cell two-electrode voltage clamp (TEVC) recording were conducted on oocytes in the presence of GABA, according to previously reported methods [17,49]. Oocytes were voltage clamped at a membrane potential of −70 mV. The oocyte recording chamber was continuously perfused with modified bath solution (MBS) with select compound (10 µM) and GABA (10 µM), with a final concentration of 1% DMSO [22,44]. It has been previously demonstrated that this concentration of DMSO did not significantly affect GABA_A_R function [49]. Oocytes were perfused at a rate of 3 mL/min. Electrophysiology recordings were conducted in triplicate compound for each compound. The change of GABA_A_R activity was determined by comparison of GABA_A_R potentiation to the EC_20_ with control saline in the presence of GABA. 

### 4.9. Pharmacokinetic and Statistical Analysis

Pharmacokinetic parameters were calculated using noncompartmental analysis. Elimination half-life (T_1/2_) was calculated with R software from the linear regression of the terminal phase of the log serum concentration-time curve. The AUC_0__→24_ was calculated with OriginLab software by trapezoidal analysis of the serum concentration-time plot. C_max_ and T_max_ were determined directly from observed data. Data are expressed as mean ± SEM. Statistical significance between AUC0_→_24 among test groups was calculated with an independent t-test (GraphPad software, San Diego, CA, USA). Time to LORR was analyzed by repeated measured ANOVA. Tolerance was determined by mean split in LORR response and confirmed with paired t-test (Week 1 vs. Week 8). Statistical significance of serum concentrations for individual time points was calculated using a two-way analysis of variance (ANOVA) analysis. Post hoc Bonferroni tests were performed when justified. Statistical significance of electrophysiology results compound GABAergic modulation was determined using a paired t-test (Vehicle control vs. Compound, GraphPad software, San Diego CA, USA). *p* < 0.05 was considered significant.

## 5. Conclusions

Our results indicate that administration route has a substantial effect on DHM bioavailability and DHM is rapidly cleared from serum and brain, correlating with the observed short duration of anti-intoxicating properties of DHM in mice. We found that IP administration of DHM led to 7.2 and 23.8 times the AUC_0__→24_ of PO administration in male and female mice, respectively. Sex was found to have a significant impact on oral bioavailability of DHM, with PO exposure levels in male mice 3.5 times that of female mice. Seven DHM metabolites were detected in serum samples, with glucuronidation, O-methylation and dehydroxylation providing for the major metabolic pathways. DHM, 3′-Me-DHM, and 4′-Me-DHM were detected in brain tissue. Electrophysiology studies in Xenopus oocytes suggest DHM may be a positive modulator and 4′-Me-DHM a negative modulator of α5β3γ2 GABA_A_Rs, although further studies are necessary to fully characterize GABAergic activity. We are the first to report DHM to significantly reduce ethanol-induced LORR in mice at the high ethanol dose of 5 g/kg PO, suggesting DHM may have utility in mitigating the effects of high alcohol levels. In conclusion, the presented work supports the potential of DHM’s mechanism at counteracting neurological effects of high doses of ethanol, suggests these effects are limited by poor absorption and rapid clearance, and highlights the need for more extensive investigation into the mechanism of DHM’s anti-intoxicating properties.

## Figures and Tables

**Figure 1 ijms-22-07460-f001:**
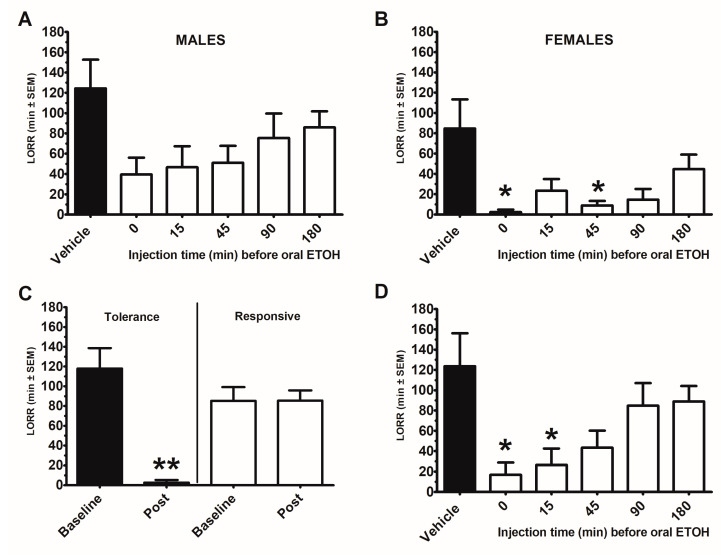
Dihydromyricetin (DHM) time course for attenuating the loss of righting reflex (LORR) following acute oral gavage of ethanol (5 g/kg) in male and female 8-week-old C57BL/6J mice. Each mouse received an IP injection of DHM or vehicle (10% DMSO) at a time prior to an oral gavage of ethanol (ETOH). Each time point (0, 15, 45, 90, 180 min) or vehicle was 1 week apart for a total of 6 weeks. (**A**) Males (*n* = 8), (**B**) Female (*n* = 8). To determine whether the observed DHM effects were a consequence of repeated oral ETOH tolerance, mice were tested for LORR responsivity prior to (Baseline) and after (Post) the 6-week repeated IP dosing schedule. For this, mice received an oral ETOH without any IP injections. Mice were divided by the significant difference in baseline and post LORR. (**C**) Mice demonstrating a significantly reduced LORR displayed tolerance (*n* = 8; 3 males/5 females), whereas mice that retained the LORR were responsive (*n* = 8; 5 males/3 females). (**D**): Time course of DHM on LORR in Responsive only mice. * indicates *p* < 0.05 from vehicle treated condition. ** indicates *p* < 0.005 from baseline.

**Figure 2 ijms-22-07460-f002:**
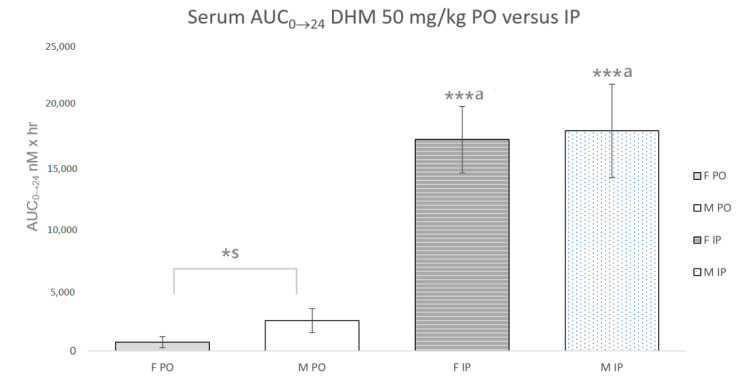
Serum AUC0_→_24 of DHM 50 mg/kg with oral (PO) and intraperitoneal (IP) administration in male and female mice. *^s^ Statistically significant (*p* < 0.05) difference between male and female mice. ***^a^ Statistically significant (*p* < 0.0001) between administration routes of same sex.

**Figure 3 ijms-22-07460-f003:**
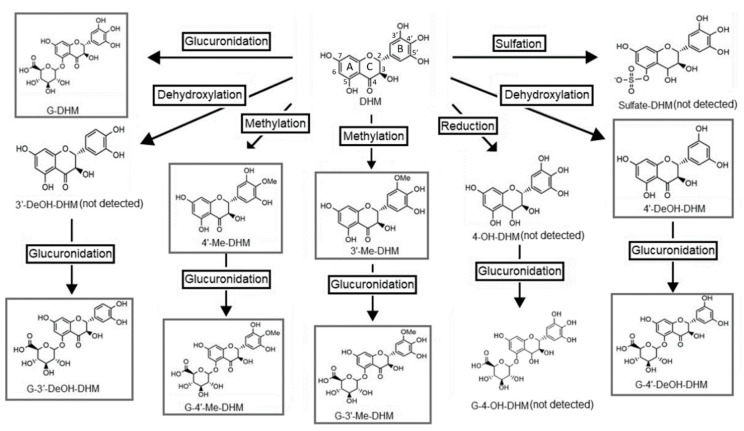
Proposed metabolic pathway of dihydromyricetin (DHM) based on detection of metabolites in feces and/or urine samples of rats [31,32]. Boxed compounds are those that were detected in this study.

**Figure 4 ijms-22-07460-f004:**
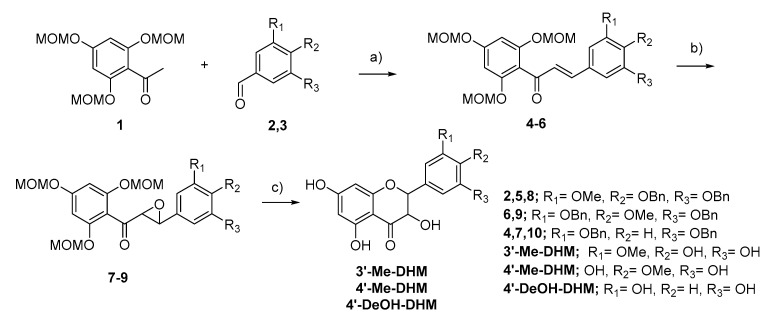
Synthesis route of 3′-Me-DHM, 4′-Me-DHM, 4′-DeOH-DHM (a) NaOH, EtOH/H_2_O (b) H_2_O_2,_ NaOH, MeOH/Dioxane (c) 1. HCl, MeOH, THF 2. (4′-Me-DHM only) Pd/C, H_2_, MeOH.

**Table 1 ijms-22-07460-t001:** Pharmacokinetic parameters, including serum elimination half-life (T_1/2_), maximum concentration (Cmax), total serum exposure (AUC0_→_24) and time of maximum serum concentration (Tmax), as well as brain concentrations (15 min post administration) of DHM 50 mg/kg PO and IP in female (F) and male (M) mice. DHM was not detected in brain samples 45 min or later post DHM administration.

		F PO	M PO	F IP	M IP
Serum	T_1/2_ (h)	1.83	1.91	0.76	1.59
	Cmax (nM)	623 ± 159	1885 ± 590	23,777 ± 1998	38,251 ± 7441
	AUC_0__→24_ (nM × h)	728 ± 461	2508 ± 988	17,360 ± 2734	18,079 ± 3840
	Tmax (min)	15	15	15	15
Brain	Concentration (nmol/g) 15 min post DHM IP	50.9 ± 2.6	423 ± 66	864 ± 220	672 ± 85

Parameters are the mean ± SEM.

**Table 2 ijms-22-07460-t002:** Intrinsic GABA_A_R activity of DHM, 4′-Me-DHM, 3′-Me-DHM and 4′-DeOH-DHM in α5β3γ2 expressed in Xenopus oocytes in the presence of GABA.

	DHM	4′-Me-DHM	3′-Me-DHM	4-DeOH-DHM
GABA_A_R Potentiation	+43.2 ± 5.6 **	Not Active	−12.6 ± 1.5 *	+5.3 ± 2.0

Parameters are the mean ± SEM. * Statistically significant (*p* ≤ 0.05), ** Statistically significant (*p* ≤ 0.01).

## Data Availability

Not applicable.

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
