# Peer review of "Identification of Dihydromyricetin and Metabolites in Serum and Brain Associated with Acute Anti-Ethanol Intoxicating Effects in Mice"

_ijms, 2021, doi:10.3390/ijms22147460_

Round 1

Reviewer 1 Report

This manuscript examines dihydromyricetin and metabolites in mouse serum and brain and tries to relate this to dihydromyricetin’s acute anti-ethanol intoxicating effects.

The manuscript has three major shortcomings:

  1. Based on the data presented, it is not possible to draw firm conclusions with regard to the effect of dihydromyricetin (DHM) and/or its metabolites on GABAA This is because a positive effect of 10 microM DHM on GABAA receptors, but the free concentration of DHM in brain extracellular fluid has not been reported. It is essential that this is fully addressed.
  2. A full PK profile of DHM in brain extracellular fluid needs to be presented.
  3. It is important that dose-response curves are presented to support the data in Table 2.

Author Response

Response to Reviewer 1 Comments

Point 1: Based on the data presented, it is not possible to draw firm conclusions with regard to the effect of dihydromyricetin (DHM) and/or its metabolites on GABAA This is because a positive effect of 10 microM DHM on GABAA receptors, but the free concentration of DHM in brain extracellular fluid has not been reported. It is essential that this is fully addressed.

Response 1: Yes, we agree and thank the reviewer for pointing this out. Based on what is known about DHM and our in vivo data, we explored the potential modulation of GABAA receptors. Our intention was to examine the time course of LORR, and DHM and metabolites bioavailability in the serum and brain. Our in vitro experiments were not intended to draw definitive conclusions, but to explore potential sites of action. As such, in the revision we have taken efforts to explain our in vitro data as a potential mechanism of action, rather than drawing firm conclusions.  

  1. In multiple locations, modifications have been made to more accurately represent the limited interpretations of the data. For example, the following changes were made:

Abstract-

“Collectively, these results suggest acute anti-intoxicating effects of dihydromyricetin with high doses of ethanol, that are limited by rapid serum and brain clearance of dihydromyricetin” has been changed to “Our results indicate that administration route and sex significantly impact DHM bioavailability in mice, with both administration routes associated with rapid serum and brain clearance. This correlates with the observed short duration of DHM’s anti-intoxicating properties and highlights the need for further investigation into mechanism of DHM’s potential anti-intoxicating properties.” (lines 26-30)

Introduction-

“Lastly, we synthesized three DHM metabolites, 3’-O-methyl-dihydromyricetin (3’-Me-DHM), 4’-O-methyl-dihydromyricetin (4’-Me-DHM), 4’-dehydroxy-dihydromyricetin (4’-DeOH-DHM), enabling preliminary screening of intrinsic α5β3γ2 GABAAR activity.” (lines 107-110)

Discussion-

“We conducted the first vitro analysis of intrinsic α5β3γ2 GABAAR activity of DHM, reporting significant positive modulation at a 10 µM concentration.” has been changed to “As a preliminary screening for GABAergic activity, we assessed DHM and 3 metabolites (10 µM) for potentiation of α5β3γ2 GABAARs expressed in Xenopus oocytes.” (lines 280-281). 

Conclusion-

“In conclusion, the presented work supports the potential of DHM’s mechanism at counteracting neurological effects of high doses of ethanol, with this effect limited by rapid brain and serum clearance and supports the specificity of DHM’s GABAAR activity” has been changed to “In conclusion, the presented work supports the potential of DHM’s mechanism at counteracting neurological effects of high doses of ethanol, suggests these effects are limited by rapid brain and serum clearance, and highlights the need for more extensive investigation into the mechanism of DHM’s anti-intoxicating properties.” (lines 498-502).

  1. In the discussion, we have expanded on the limitations in interpretation of DHM’s GABAergic activity to add the following text:

“While the observed in vitro GABAergic activity of DHM and 3’-Me-DHM is significant, it is not conclusive of in vivo GABAAR activity. Since this activity is limited to a 10 µM concentration and we did not determine free extracellular brain levels, we cannot conclude that compounds reach levels consistent with the reported GABAergic activity. This is a limitation of this study that highlights the need for further investigations into GABAergic activity of DHM and metabolites.” (lines 290-295).

Point 2: A full PK profile of DHM in brain extracellular fluid needs to be presented.

Response 2: We agree with the reviewer that further clarification of this point was necessary.

We have expanded our discussion to clarify the major limitations in interpretation of the reported brain tissue concentrations of DHM and metabolites. We have expanded upon the following text.

“Further, it is important to note that brain tissue levels do not accurately represent free CSF levels of xenobiotic, necessary to exert effect. Thus, interpretation of future analysis of cerebral spinal fluid (CSF), could provide for a more accurate representation of the relationship between DHM and bioactivity.” (lines 269-273)

Point 3: It is important that dose-response curves are presented to support the data in Table 2.

Response 3: As stated, one of the major limitations to the interpretation of our in vitro findings, is that we did not perform a dose-response curve of DHM. Our rationale was to use this in vitro assay as a screening tool for the action of DHM metabolites, 4’-Me-DHM, 3’-Me-DHM, and 4-DeOH-DHM, on GABA potentiation of α5β3γ2 GABAAR activity to strengthen our in vivo findings. The doses of DHM and GABA used in our present study were based on previous studies (Shen et al., Journal of Neuroscience, 2012) and studies using similarly structured flavonoids (Hanrahan et al, British Journal of Pharmacology; 2011), we added these citations to the methods. Future studies are planned to provide a more comprehensive analysis of DHM metabolites on GABA receptors and subtypes. We have expanded upon the following text.

“While interpretation of the reported GABAergic activity is limited, previous studies support the reliance of DHM’s anti-intoxicating properties on GABAergic activity. For instance, in rats, the anti-alcohol effects of DHM were antagonized by flumazenil, a BZD antagonist, in a dose-dependent manner [22]. This strongly supports the reliance of DHM’s acute anti-intoxicating properties on specific GABAAR interactions in vivo. There are currently no previous reports of GABAergic activity of 3’-Me-DHM, suggesting a need for further investigation. However, is improbable is that 3’-Me-DHM plays a major role in a GABAAR-mediated mechanism of DHM, as methylated metabolites were only minimally detected in brain tissue. Thus, unless extremely potent, it is unlikely that 3’-Me-DHM reaches levels necessary to exert prolonged GABAergic activity. Notably, the selective GABAAR positive modulation by DHM correlates with reported SAR studies surrounding flavone GABAAR activity that suggest minor molecular modifications significantly influence positive versus negative GABAAR modulation [42-45]. This also further supports the specificity of DHM’s GABAAR activity, necessitating further investigation to more fully characterize the GABAAR activity of DHM and metabolites.” (lines 296-310)

Reviewer 2 Report

The study reported by carry et al. investigated the mechanism for the treatment of alcohol use disorder by a natural flavonoid “Dihydromyricetin” in C57BL/6J mice. Dihydromyricetin (50 mg/kg IP) administered 0 or 15-min prior to ethanol (PO 5 g/kg) significantly reduced ethanol-induced loss of righting reflex. 

Total serum exposures (AUC0→24) of dihydromyricetin through intraperitoneal (IP) administration led to 23.8-fold and 7.2- increases in male and female mice. Electrophysiology studies were done in Xenopus Oocytes demonstrated that dihydromyricetin (10 μM) could potentiate GABAergic activity (+43.2%), and the metabolite, 4-O- methyl-dihydromyricetin (10 μM), to negatively modulate GABAergic activity (-12.6%). The result suggested that dihydromyricetin may have acute anti-intoxicating effects in high doses of ethanol. Some points to be considered for the betterment of the manuscript.

Decision: Minor decision 

Below are the comments for this paper to be incorporated in the revised version of the manuscript. 

  1. Line 24 Why it was “Xenopus Oocytes” italic, I found inside the manuscript the authors used the word in normal format, please consider it same in whole MS. Line 201 (please check)
  2. Line 37 “This contrasts” What does the author mean
  3. Line 41-42 please mention the information in a simple language, whether it inhibit or activate in the development of AUD
  4. Line 46-47 Further supporting this, BZDs and ethanol both lead to the plasticity of GABAARs
  5. in-vivo anti-ethanol effects antagonized by the addition of flumazenil, a BZD antagonist [18, 19]. Did the author checked with DHM
  6. Line 85, in vivo sometimes author used in italic or some time in normal form. Please be consistent about it (line 237)
  7. Line 136 The subtitle should be change
  8. Line 222, why there was author name and year, other did not
  9. Line 233 influenced by P-gp expression. What does it mean?
  10. Line 253-254 what would be the reason
  11. Line 267, missing in (in vitro)
  12. Line 299-301 please explore it for clear meaning
  13. Line 466, please check the line
  14. Line 578 there was no number as similar to the above
  15. Almost all references needed to be modified based on IJMS journal.

Author Response

Point 1: Line 24 Why it was “Xenopus Oocytes” italic, I found inside the manuscript the authors used the word in normal format, please consider it same in whole MS. Line 201 (please check)

Response 1: Thank you for pointing this out. The italics were in error and we have changed all text accordingly.

Point 2: Line 37 “This contrasts” What does the author mean

Response 2: Thank you for pointing out the lack of clarity in this sentence. Line 36-50 have been changed accordingly, to more represent the intended meaning more accurately.

Point 3: Line 41-42 please mention the information in a simple language, whether it inhibit or activate in the development of AUD.

Response 3: Thank you for clarifying the need to simplify the language (lines 51-54)

Point 4: Line 46-47 Further supporting this, BZDs and ethanol both lead to the plasticity of GABAARs

Response 4: We have also simplified the language (lines 54-59).

Point 5: in-vivo anti-ethanol effects antagonized by the addition of flumazenil, a BZD antagonist [18, 19]. Did the author checked with DHM

Response 5: Thank you for pointing out the need to clarify this. Yes, the in vivo anti-intoxicating effects of DHM were antagonized by flumazenil in a dose-dependent manner. The text has been modified to clarify this (lines 72-74).

Point 6: Line 85, in vivo sometimes author used in italic or some time in normal form. Please be consistent about it (line 237)

Response 6: Thank you for pointing this out. The text has been modified accordingly.

Point 7: Line 136 The subtitle should be change

Response 7: The text has been modified accordingly (line 145).

Point 8: Line 222, why there was author name and year, other did not

Response 8: The text has been modified to IJMS format.

Point 9: Line 233 influenced by P-gp expression. What does it mean?

Response 9: We have expanded upon this to clarify the effect rodent species and sex may have on Pgp expression and DHM bioavailability (lines 238-244) to read:

“This suggests that Pgp efflux plays a significant role in reducing DHM absorption and correlates with reports of DHM to act asa Pgp substate and inhibitor [36, 37]. Sex was also found to have a significant influence on DHM serum exposure following PO, but not IP administration. As prior studies suggest that sex and certain hormones can impact Pgp expression [38, 39], variations in Pgp expression between sexes might contribute to differences in DHM PO exposure.”

Point 10: Line 253-254 what would be the reason.

Response 10: Thank you for pointing out the need to expand on this (lines 256-259) to read:

“The high fecal concentration of this 4’-DeOH-DHM, coupled with minimal serum levels, correlates with this metabolite being produced in the large intestines by bacterial fermentation [40] but is poorly absorbed into the blood stream”.

Point 11: Line 267, missing in (in vitro).

Response 11: We have modified the text accordingly.

Point 12: Line 299-301 please explore it for clear meaning.

Response 12: We have modified the text to clarify this (lines 304-310).

Point 13: Line 466, please check the line.

Response 13: The line numbers have been fixed accordingly.  

Point 14: Line 578 there was no number as similar to the above.

Response 14: The line numbers have been fixed accordingly.

Point 15: Almost all references needed to be modified based on IJMS journal.

Response 15: Thank you for pointing this out. The references are now updated following IJMS guidelines.

Round 2

Reviewer 1 Report

The authors have addressed my concerns, but a few issues remain:

The authors refer to future studies to measure DHM concentration in CSF, which is fine. However, a much more direct measurement would be DHM concentration in brain extracellular fluid using tissue microdialysis. Also, line 272 should include the word 'concentration' after 'DHM'. This needs to  be addressed.

Also, line 500: the concentration of DHM in blood is not just mediated by clearance (metabolism and elimination), absorption plays a role too. This needs to be corrected, along with the suggestion of DHM metabolism occurring in the brain. I am not sure if it does or doesn’t, but I suspect that brain concentrations will reflect DHM concentration in blood. This needs to be clarified and the  possible influence of PgP transport on DHM concentrations in brain extracellular fluid. mentioned.

Author Response

Response to Reviewer 1 Comments Round 2

Point 1: The authors refer to future studies to measure DHM concentration in CSF, which is fine. However, a much more direct measurement would be DHM concentration in brain extracellular fluid using tissue microdialysis. Also, line 272 should include the word 'concentration' after 'DHM'. This needs to be addressed.

Response 1: Thank you for pointing this out. We have modified the text, adding the following sentence (lines 271-275).

“Further, it is important to note that brain tissue levels do not accurately represent free extracellular brain levels of xenobiotic, necessary to exert effect. Thus, future analysis of brain extracellular fluid using tissue microdialysis could provide for a more accurate representation of in vivo GABAAR exposure to DHM.”

Point 2: Also, line 500: the concentration of DHM in blood is not just mediated by clearance (metabolism and elimination), absorption plays a role too. This needs to be corrected, along with the suggestion of DHM metabolism occurring in the brain. I am not sure if it does or doesn’t, but I suspect that brain concentrations will reflect DHM concentration in blood. This needs to be clarified and the role of Pgp efflux on brain exposure to xenobiotics.

Response 2: We agree and thank you for pointing this out. We have modified the text in the abstract (line 27-28) and conclusion (line 500-502) to clarify that absorption plays a role in DHM bioavailability. We have also modified the discussion to mention the role of Pgp efflux on xenobiotic brain exposure (line 246-266). As you mentioned, we do not have conclusive evidence that DHM is metabolized in the brain and have deleted the sentence in the text.

Abstract

The following text has been modified (lines 27-28)

“Our results indicate that administration route and sex significantly impact DHM bioavailability in mice, which is limited by poor absorption and rapid clearance. This correlates with the observed short duration of DHM’s anti-intoxicating properties and highlights the need for further investigation into mechanism of DHM’s potential anti-intoxicating properties.”

Discussion

The following sentence has been added (lines 264-266)

“Further, as Pgp transport plays a significant role in the efflux of xenobiotics from the BBB, it is likely that Pgp efflux contributes to the rapid brain clearance of DHM.”

The following text has been modified (lines 262-264)

“We detected DHM 15 min post PO and IP administration of DHM, but not at 45 minutes later.  As brain exposure of xenobiotics is dependent on blood concentrations, this correlates with the observed rapid DHM serum clearance.”

Conclusion

The following text has been modified (lines 500-502)

In conclusion, the presented work supports the potential of DHM’s mechanism at counteracting neurological effects of high doses of ethanol, suggests these effects are limited by poor absorption, and rapid clearance, and highlights the need for more extensive investigation into the mechanism of DHM’s anti-intoxicating properties.
